# Microstructural Characterization of Laser Weld of Hot-Stamped Al-Si Coated 22MnB5 and Modification of Weld Properties by Hybrid Welding

**DOI:** 10.3390/ma14143943

**Published:** 2021-07-14

**Authors:** Hana Šebestová, Petr Horník, Šárka Mikmeková, Libor Mrňa, Pavel Doležal, Jan Novotný

**Affiliations:** 1Institute of Scientific Instruments of the Czech Academy of Sciences, Královopolská, 147, 61264 Brno, Czech Republic; hornik@isibrno.cz (P.H.); sarka@isibrno.cz (Š.M.); mrna@isibrno.cz (L.M.); jann@isibrno.cz (J.N.); 2Institute of Materials Science and Engineering, Faculty of Mechanical Engineering, Brno University of Technology, Technická, 2896/2, 61669 Brno, Czech Republic; dolezal@fme.vutbr.cz

**Keywords:** 22MnB5, laser welding, hybrid welding, cooling rate, tensile strength, microstructure, microhardness

## Abstract

The presence of Al-Si coating on 22MnB5 leads to the formation of large ferritic bands in the dominantly martensitic microstructure of butt laser welds. Rapid cooling of laser weld metal is responsible for insufficient diffusion of coating elements into the steel and incomplete homogenization of weld fusion zone. The Al-rich regions promote the formation of ferritic solid solution. Soft ferritic bands cause weld joint weakening. Laser welds reached only 64% of base metal’s ultimate tensile strength, and they always fractured in the fusion zone during the tensile tests. We implemented hybrid laser-TIG welding technology to reduce weld cooling rate by the addition of heat of the arc. The effect of arc current on weld microstructure and mechanical properties was investigated. Thanks to the slower cooling, the large ferritic bands were eliminated. The hybrid welds reached greater ultimate tensile strength compared to laser welds. The location of the fracture moved from the fusion zone to a tempered heat-affected zone characterized by a drop in microhardness. The minimum of microhardness was independent of heat input in this region.

## 1. Introduction

Press-hardened ultra-high-strength steels are excellent materials for lightweight design, especially in the automotive industry. Hot-stamped 22MnB5 has an exceptional tensile strength of about 1500 MPa and fatigue strength of about 600 MPa [1]. These properties make it possible to achieve 30−50% weight savings of a final component compared to conventional cold-worked steels. This steel is particularly well suited for automobile structural and safety components requiring good crash resistance, e.g., B-pillars, door, floor, and roof reinforcements, dash panel cross members and roof members, or front and rear bumper beams.

The process of hot stamping has two phases. First, the original sheet is austenitized at 900–950 °C for about 5–7 min. Then, it is transferred into a tool, where it is simultaneously formed and quenched by a cooled die. The great tensile strength in a quenched state results from a small amount of boron in the steel. The hardenability of steel is much more increased by boron compared to the effect of chromium, molybdenum, or vanadium. The boron has more than a hundred times greater hardening effect than other alloying elements and its content is typically only a few thousandths of weight percent [2]. The boron delays a diffusion-controlled transformation of austenite and promotes the formation of martensite [3].

Surface coatings are used to prevent sheet surface oxidation and decarburization during hot stamping. Taylor and Clough [4] refer that most austenitization furnaces operate in an inert gas atmosphere of nitrogen, hydrogen, and/or argon. Nevertheless, the austenitized blank is exposed to the atmospheric air during the transport from the furnace to the hot-forming tool. Thus, coated sheet steels are favorable even when a furnace operating with inert gas is utilized. The most common coating consists of approximately 90% aluminum (Al) and 10% silicon (Si). The melting point of Al-Si coating is only about 600 °C, which is lower than the austenitization temperature required during hot stamping. Grauer et al. [5] demonstrated that liquefaction of coating accelerates the diffusion of iron (Fe) into Al-Si, resulting in the formation of Al-Si-Fe intermetallics with high melting points. It re-solidifies quickly and maintains the protective purpose of the coating. Windmann et al. [6] found that the transformation of Al-Si coating into intermetallic phases is inhibited by silicon.

Hot-stamped components need to be joined together to produce complex shapes, and 22MnB5 has good spot weldability for both similar and dissimilar welds. The structure of the resistance spot welded joint between Al-Si coated 22MnB5 and zinc (Zn) coated H340LAD was investigated by Sejč et al. [7]. Laser welding is also a suitable technology for press-hardened steel welding. Gu et al. [8] studied a hot-stamped steel equivalent to 22MnB5. They described the effect of welding speed on microstructure and properties of butt joints performed with Nd:YAG laser. They pointed out the heat-affected zone (HAZ) softening caused by the tempering of the original martensitic microstructure. The width of the tempered zone decreased with increasing welding speed, thus with decreasing heat input. Tensile test specimens fractured in the softened zone. Similar results were achieved by Jia et al. [9] for 22MnB5 fiber laser butt welds. They also reported that the fatigue life of welded joints was lower compared to the base metal (BM).

Laser welding of coated 22MnB5 is somewhat more challenging. Lin et al. [10] investigated the effect of Al-Si coating presence on 22MnB5 laser weld microstructure and properties before and after hot stamping. They found that the tensile strength of coated and de-coated sheets in a laser-welded condition is about the same and comparable with that of the BM. The subsequent hot stamping heavily reduces the strength of coated sheets’ joints compared to the hot-stamped de-coated sheets’ joints. 

Wang et al. [11] studied the effect of welding atmosphere on the microstructure of Al-Si coated 22MnB5 laser lap joints. They found that the amount of oxygen and nitrogen in the fusion zone (FZ) is not significantly higher in the case of welds produced in the air atmosphere compared to the argon atmosphere. No welding defects were observed even in the case of the air atmosphere. Aluminum oxides less than 1 μm did not degrade weld properties. The argon atmosphere resulted in a higher content of delta ferrite in FZ. 

He et al. [12] experimentally studied the effect of laser power, welding speed, and coating condition on the microstructure and mechanical properties of 22MnB5 lap joints. Increasing laser power and decreasing welding speed led to the increase in shear strength. They observed Al-rich curved bands along the fusion line (FL) of samples with coating. They supposed them to be Fe-Al intermetallic phases responsible for the decrease in shear strength. Kim et al. [13] compared the properties of fiber laser overlap joints of 22MnB5 hot-formed steel with and without Al-Si coating. Although the HAZ softening was also observed, the samples without coating fractured at the interfacial surface during shear tensile tests. The samples with coating fractured along the FL as a result of the intermetallic Fe-Al phase developed during the welding. However, the presence of the intermetallic phase with 3 wt.% of aluminum in a dominantly Fe-Al-Si system is questionable. Kügler et al. [14] mentioned Al-rich agglomerates along with the FL of 22MnB5 laser lap joint. They contained about 7% of Al, while only 2% of Al was found in the FZ. The tensile test specimens fractured at the FL in the connection area of sheets. The wider connection area resulted in higher tensile strength. Norman et al. [15] used hybrid laser-arc technology with a filler wire to weld the 22MnB5 with and without coating. They monitored the welding process with a high-speed camera and studied the formation of oxides on the melt pool surface affecting the dilution of a melt pool, leading to the formation of Al-rich intermetallic phases.

Sun et al. [16] identified Al-rich regions as delta ferrite and proposed the use of Ni interlayer to reduce its fraction [17]. Lee et al. [18] proposed high-carbon filler wire to prevent a brittle fracture of press-hardened laser welds of Al-Si coated 22MnB5. A similar approach was applied by Lin et al. [19] to increase the tensile strength of welded joint in a hot-stamped condition. The sheet surface could be also mechanically treated or laser-ablated [20] before the welding to remove the protective coating to forestall its negative effect on strength of laser welds. Nevertheless, these procedures prolong the time of component production and thus increase its final price.

Our research aims to investigate the microstructure and mechanical properties of butt laser welds of thin hot-stamped Al-Si coated 22MnB5 sheets. Phase analysis utilizing energy-dispersive X-ray spectroscopy (EDS) and electron backscatter diffraction (EBSD) is used to identify the microstructure of the coating and weld fusion zone, especially to confirm the presence of ferrite. We propose a hybrid technology of welding combining laser and tungsten inert gas (TIG) welding without a filler wire to modify weld metal microstructure to increase the tensile strength of welds. The effect of increasing heat input on weld macrostructure, tensile strength, and microhardness is investigated.

## 2. Materials and Methods

### 2.1. Welding Conditions

The IPG YLS2000 fiber laser (IPG Photonics, Oxford, MS, USA) with a maximum output of 2 kW was used for butt welding experiments. The laser beam was delivered by a 200 µm optical fiber to the Precitec YW30 processing head. The head was mounted on the arm of the ABB IRB 2400 robot. The beam was focused with the lens with a 200 mm focal length. The focal point was 1 mm under the metal sheet surface. The spot diameter reached 0.4 mm on the sheet surface. Besides the laser welding, a hybrid technology of laser-TIG welding in laser-leading configuration was experimentally investigated. We described this technology in detail in Šebestová et al. [21]. The Fronius MagicWave 1700 Job (Fronius International, Wels, Austria) tungsten inert gas welding source was employed to reduce the weld metal cooling rate and to improve weld surface properties. The tungsten electrode with a 2.4 mm diameter was fixed with a special holder to the laser welding head to be 3 mm above the workpiece. It was tilted off 45 degrees and distanced 3 mm towards the beam axis. The scheme of experimental configuration is depicted in Figure 1.

The experimental material 22MnB5 was supplied by ArcelorMittal with the trade name Usibor 1500. It was covered with protective Al-Si coating AS150, i.e., 150 g.m^−2^. The metal sheets were delivered after hot stamping in a flat die. Their dimensions were 200 mm × 300 mm. The thickness of a sheet varied from 1.6 mm to 1.9 mm. Pieces to be welded were degreased, fixed in mounting jig, and tack welded before the seam welding itself.

Laser power 1 kW, welding speed 20 mm⋅s^−1^, and pressure of argon shielding gas 18 l⋅min^−1^ were kept constant during all experiments. The only variable was the welding current of the TIG source. It reached 0 A, 20 A, 40 A, and 60 A. The 0 A represents the laser welding. The higher values characterize the hybrid technology. The current 20 A was the minimum current at which a stable arc was achieved at 20 mm⋅s^−1^, which is a relatively high welding speed for TIG welding. The effect of increasing arc current is evaluated. The values above 60 A represent the arc supply energy larger than that of the laser beam and lead to undesirable widening of HAZ. The TIG welding was performed in a direct current electrode negative (DCEN) regime.

### 2.2. Methods of Weld Inspections

The welds were sectioned perpendicularly to the welding direction. The weld cross-sections were fixed in epoxy resin. They were ground and polished up to 0.25 μm. Finally, the samples for optical microscopy were etched with 3% Nital for 8 s to reveal their macro and microstructure. The samples for electron microscopy were polished electrolytically in a mixture of acetic and perchloric acid (19:1) at 20 V for 4 s.

The chemical composition of the BM was analyzed by glow discharge optical emission spectroscopy (GDOES) using Spectrumat GDS 750 (Spectruma Analytik GmbH, Hof, Germany).

Weld macrostructure was examined with the Olympus SZ61 stereo-microscope (Olympus, Tokio, Japan). The Keyence VK-X laser scanning confocal microscope was used for observation of weld microstructure at magnifications up to 3600×. The scanning electron microscope (SEM) Magellan 400 (FEI Company, Hillsboro, OR, USA) with the submicron resolution was used for higher magnifications. The EDAX Octane Elect Super Silicon Drift Detector and EDAX Hikari (both AMETEK EDAX, Mahwah, NJ, USA) were used for EDS and EBSD, respectively. The EDS analysis was performed at 20 keV primary beam energy and 1.6 nA beam current. The spot acquisition time was 1 minute for each point. The EBSD data were collected at 20 keV landing energy of the primary electrons and beam current of 3.2 nA. The step size of 20 nm was utilized to obtain a high quality of data.

LECO LM 247 AT (LECO Corporation, St. Joseph, MO, USA) was used to evaluate the Vickers microhardness according to the ISO 6507-1 and ISO 9015-2 standards. The 0.98 N load was applied for 10 seconds. The measurement across each weld was performed 1 mm under the face surface. The indents were spaced 0.1 mm.

The tensile characteristics were measured according to the ISO 6892-1 and ISO 4136 standards [22,23]. The tensile tester ZD 10/90 (FRITZ HECKERT, Chemnitz, Germany) equipped with the MFA extensometer was used. The specimens were laser-cut with nitrogen processing gas. The original gauge length and parallel length of test specimens were 80 mm and 100 mm, respectively. The width of a parallel length and the width of a grip section was 25 mm and 37 mm, respectively. The total length was 250 mm.

## 3. Experimental results

Both the BM and weld microstructures and mechanical properties were analyzed.

### 3.1. Base Metal Properties

The average chemical composition of studied steel measured by GDOES is concluded in Table 1.

The microstructure of the as-received 22MnB5 is ferritic-pearlitic. However, the subsequent hot stamping results in the martensitic microstructure. The microstructure of the BM and the detail of surface coating after hot stamping are shown in Figure 2. The coatings are in the range of 30–45 μm. Large mechanical and thermal stresses present during the hot stamping process promote multiple brittle fractures of the coating and its delamination.

The coating is not homogeneous due to the diffusion present during hot stamping. The EBSD analysis revealed the crystallographic structure of the coating subzones. They have also a good contrast in the confocal image (Figure 2b). The inverse pole figure (IPF) map and the corresponding phase and image quality (IQ) map of the coating–steel interface are shown in Figure 3. Besides the phases mentioned in Figure 3, the presence of gamma iron, Al_8_Fe_2_Si, and Al_9_Fe_2_Si_2_ was investigated. However, these phases were not found.

The chemical differences between the BM and coating subzones were investigated by the EDS line scan (along the dark green line) and spot measurements (Figure 4). The martensitic microstructure of the BM (spots 1−3) is followed by a ferritic zone very rich in aluminum and silicon (spot 4). The solubility of silicon in alpha-iron is up to about 18% [24]. Based on the EBSD analysis, the following and the widest zone is formed dominantly by intermetallic phase Al_5_Fe_2_ (spots 5−6). The last visible zone in EBSD images is the ferritic solid solution again (spots 7−8). However, the content of Al is higher compared to the zone adjacent to the steel surface.

### 3.2. Weld Macro- and Microstructure

#### 3.2.1. Weld Macrostructure

The macrostructures of laser weld and hybrid welds are shown in Figure 5. All the welds are free of pores or cracks. Nevertheless, the following imperfections occur. Laser weld and hybrid welds made with 20 A and 40 A have incompletely filled grooves which is typical when welding without a filler. The weld sagging does not exceed 0.2 mm. Therefore, these welds could reach weld quality level B according to ISO 13919-1 [25]. The lack of root fusion occurred in the case of laser weld and 60 A hybrid weld. Its depth is almost 0.5 mm. This is quite a serious defect impeding the weld to reach the lowest quality level D. Therefore, the only welds fulfilling the criteria of B level are hybrid welds made with 20 A and 40 A. It seems that there is a both-sided undercut in hybrid weld made with 40 A. Nevertheless, it has been refuted during the observation with the confocal microscope at higher magnifications. These dark-appearing regions in a stereomicroscope image in Figure 5 appear white in a confocal image in Figure 6. Figure 6 represents the area marked with a yellow rectangle in Figure 5. It is obvious that this region is a part of FZ, not a geometrical defect. Its etching shade is similar to the etching shade of protective coating. Therefore, a high concentration of aluminum and silicon limiting the martensitic transformation can be expected in this region of FZ.

Three cross-sections of each weld were analyzed. Figure 7 presents the average cross-section areas of FZ and surrounding HAZ visible on macrographs. The error bars represent standard deviations. Both the FZ and HAZ broaden with increasing arc current. However, the increase in HAZ cross-section area is higher compared to FZ.

#### 3.2.2. Fusion Zone Microstructure

The microstructure of both the FZ and HAZ depends on a time-temperature history of a specific weld region. The FZs of all welds have casting microstructure with columnar grains of prior austenite oriented in the direction of the most intense heat transfer during cooling. The FZ was investigated along the weld axis, about 0.5 mm under the surface of each weld face side. The comparison of FZ of laser weld and hybrid weld made with 60 A is shown in Figure 8. The microstructure of all welds is quite similar and dominantly formed by lath martensite.

Besides martensite, the laser weld exhibits large white-etching curved bands. These regions are observable in hybrid welds, as well. However, their size is much smaller and their distribution is more uniform. EBSD and EDS analysis were performed to identify their microstructure and composition. Figure 9 presents the IPF map combined with the IQ map and the phase map of a part of laser weld FZ. Based on the crystallographic lattice characteristics, the EBSD conclusively proved the presence of large elongated ferritic areas within the martensitic FZ (Figure 9). Less than 1% of retained austenite was discovered in FZ. The content of Al and Si in ferritic areas was 3.9 wt.% and 1.1 wt.%, respectively, while 2.8 wt.% and 0.9 wt.%, respectively, were detected in martensite.

#### 3.2.3. Heat-Affected Zone Microstructure

There are several subzones of HAZs visible in Figure 5. Their widths differ based on the variation of heat input. However, the microstructure of the corresponding subzones is the same. Figure 10 presents the coarse-grained (CGHAZ), fine-grained (FGHAZ), intercritical (ICHAZ), and subcritical (SCHAZ) part of HAZ of the hybrid weld made with 20 A. 

The gradual decrease in peak temperature (*T*_p_) from FL towards BM results in rather a gradient character of CGHAZ (*T*_p_ >> *A*_c3_) and FGHAZ (*T*_p_ > *A*_c3_). Both regions are re-austenitized and formed by a newly produced martensite. The ICHAZ (*A*_c1_ < *T*_p_ < *A*_c3_) is characterized by only a partial re-austenitization of the BM resulting in a structure combining newly formed martensite and a tempered original structure. This zone has good contrast in an optical image (Figure 5). The SCHAZ (*T*_p_ < *A*_c1_) is composed of tempered martensite. The degree of tempering gradually decreases with decreasing *T*_p_. The SCHAZ/BM interface is difficult to distinguish by optical microscopy. The image in Figure 10d was taken near the ICHAZ/SCHAZ interface.

### 3.3. Microhardness Measurement

The average BM microhardness is (478 ± 19) HV0.1. The profiles of microhardness measured across weld cross-sections 1 mm under the face side are displayed in Figure 11. The BM is being tempered when the *T*_p_ rises until the *A*_c1_ temperature is reached. The maximal level of tempering corresponds to the lowest value of microhardness. It drops under about 300 HV0.1 at the interface of SCHAZ and ICHAZ.

Further increase in *T*_p_ results in partial and at higher temperatures in total austenitization of the HAZ. During the cooling, the austenite transforms into martensite and the microhardness rises again. The maximal average microhardness is found in FGHAZ and it increases with increasing heat input. The CGHAZ reaches about 2−6% lower values. Similarly, the average microhardness of FZ increases with increasing heat input. However, according to the standard deviation, it is comparable with the BM microhardness in the case of hybrid welds. About a 6% decrease in FZ microhardness towards the BM microhardness is identified in laser weld.

Both the width of the surface coating and the dimensions of white-appearing regions are relatively small. Therefore, the load had to be reduced to investigate their microhardness. The surface coating after hot stamping is hard and brittle and reaches (1065 ± 40) HV0.025. The microhardness of the ferritic region in FZ of laser weld is only (270 ± 12) HV0.025. The positions and appearance of indents are visible in Figure 12.

### 3.4. Strength Properties Measurement

The tensile properties of both the BM and the welded samples were investigated (Figure 13). The ultimate tensile strength (UTS) and the yield strength of the BM before the hot stamping reached (603 ± 1) MPa and (438 ± 1) MPa, respectively. After the hot stamping, it increased to (1375 ± 18) MPa and (1217 ± 17) MPa, respectively. The welding led to a decrease in tensile properties. The UTS of laser welds is the lowest and these welds fractured in the FZ during the tensile test. On the other hand, all the samples welded with hybrid technology broke close to the interface of visible HAZ and BM, i.e., in the SCHAZ. The UTS of laser weld and hybrid welds made with 20 A, 40 A, and 60 A corresponds to 64% and 89%, 85%, and 82%, respectively, of the UTS of hot-stamped BM.

The total elongation at fracture *A*_t_ of the original BM is (18 ± 1)%. After the hot stamping, it is reduced to (2.9 ± 0.4)%. The elongation of welded hot-stamped 22MnB5 dramatically drops under the value of 1% for all tested heat inputs.

## 4. Discussion

Laser welds always fractured in the FZ during tensile tests. This is consistent with the results of Lin et al. [10] and other researchers. Laser welds reached only 64% of the BM’s UTS. Large continuous curved bands of a secondary phase within the dominant martensitic FZ are responsible for the drop in laser weld strength. The FZ itself is enriched with ferrite-forming elements originating from the Al-Si coating of the BM. The secondary phase has about 1% higher content of Al compared to the martensitic matrix. Similar results were reached by He et al. [12] and Kim et al. [13], who observed curved bands of a minor phase close to FL. Based on a higher content of Al, they identified it as the Fe-Al intermetallic phase. However, the solubility of Al in Fe is much higher than the 4% found in these bands [26]. Our EBDS analysis discovered a body-centered cubic lattice of iron in these bands. Thus, it is evident that they are formed by a ferritic solid solution, not an intermetallic phase. This result is also consistent with microhardness measurements. The ferritic bands are soft with about 270 HV0.025. On the other hand, the Al_5_Fe_2_ intermetallic phase detected in the coating is brittle and has almost four times higher microhardness. Nanoindentation measurements of Saha et al. [27] confirmed much lower hardness of ferritic regions compared to the dominant martensitic phase.

The ferritic islands were found also in hybrid welds. Nevertheless, they are much smaller, non-continuous, and uniformly distributed throughout the FZ. Therefore, they do not have such a negative effect on weld strength properties. Hybrid welds reached higher UTS compared to laser welds. The location of fracture of hybrid weld joints during the tensile tests was close to the interface of visible HAZ and BM, i.e., in the SCHAZ. This area is characterized by a drop in microhardness caused by the tempering of the original martensitic microstructure during the welding. The degree of softening is not higher at higher arc currents. The average value is (300 ± 12) HV0.1. However, the width of the tempered zone increases with increasing arc current, resulting in the decrease in UTS.

The difference between the size and distribution of ferritic islands in laser and hybrid welds results from dissimilar cooling conditions of a melt pool. The laser beam supplied energy (a ratio of beam power and welding speed) is 50 J⋅mm^−1^. The additional arc energy (product of arc voltage and arc current divided by welding speed) is 25 J⋅mm^−1^, 38 J⋅mm^−1^, and 48 J⋅mm^−1^ for 20 A, 40 A, and 60 A, respectively. Although the absorption efficiency of laser and arc energy is different, increasing arc current always increases the overall heat input. Based on our finite element method simulations of 3 mm sheet but welding, the additional energy of 40 A arc can reduce the cooling rate of FZ by about 40−50% depending on material composition.

In our set of experiments, laser welding is characterized by the lowest heat input and thus the fastest cooling of a melt pool. Although the surface coating of the BM dissolves during steel melting, the rapid cooling rate accompanying laser welding does not allow sufficient stirring and homogenization of the melt pool. The insufficient diffusion of ferrite-forming chemical elements of the coating leads to the formation of large ferritic bands weakening the laser weld joint. The reduced cooling rate of hybrid welding supports the homogenization of a melt pool which promotes the formation of smaller islands of ferrite. The martensite is still the dominant microstructure of hybrid welds. It is consistent with the continuous cooling diagram of 22MnB5 presented by Taylor and Clough [4].

Increasing heat input usually increases the degree of softening in martensitic steel [16]. In our set of experiments, heat input variations were relatively low. Therefore, a similar holding time can be expected in the SCHAZ resulting in the same degree of softening.

The microhardness of FZ of hybrid welds made with 40 A and 60 A was higher compared to laser weld. This is somewhat surprising because higher heat input leads to a larger volume of a melt pool and thus a lower cooling rate. However, the cooling rate is not the only variable. These welds are characterized by a wider FZ. Therefore, the volume of melted coating increases. The portion of FZ width and FZ cross-section area increases with increasing current. Thus, a higher content of silicon originating from the coating can be present in the FZ at higher heat inputs. Silicon strengthens the ferrite and increases the hardenability of austenite [28,29]. Therefore, a higher content of silicon could be responsible for the FZ hardness increase. Nevertheless, these assumptions must be confirmed yet. About 1 wt.% of Si was found in laser welds.

The cooling rate of autonomous laser welding can also be reduced by lowering the welding speed. However, it leads to the production efficiency decrease. On the other hand, the increase in laser power can be responsible for welding defects such as face and root depression worsening. On the other hand, autonomous TIG welding is characterized by a high heat input, resulting in slower cooling. However, this technology is not suitable for hot-stamped components welding because it produces wide HAZ with degraded microstructure. The desired martensitic microstructure achieved by hot stamping is tempered in a wide area which leads to strength loss. The production efficiency would also be smaller compared to laser welding since lower welding speeds are applied in arc welding. The modification of the weld cooling rate via the addition of arc to laser technology is effective. This hybrid technology is easy to implement and does not represent significant investment costs. Primarily, the relatively high welding speed can be preserved. However, an optimal combination of processing parameters must always be found. The TIG implementation also allows increasing the overall heat input when the maximal laser power is insufficient for the full penetration. It also facilitates bridging the gap between sheets to be welded.

## 5. Conclusions

Large, curved Al-rich ferritic bands were identified in the FZ of laser welds of hot-stamped Al-Si coated 22MnB5. These bands are responsible for a drop in UTS. The diffusion of coating elements is inhibited during rapid heating and cooling of laser welding. The melt pool is not sufficiently homogenized before it solidifies.

The heat of arc employed in hybrid laser-TIG technology reduces the weld cooling rate and thus prolongs the time for weld metal homogenization. The ferritic islands are also formed. However, they are much finer and more evenly distributed throughout the FZ.

All hybrid welds fractured in a tempered SCHAZ during the tensile test and reached higher UTS than the laser weld fracturing in the FZ. The lowest applied arc current 20 A resulted in the highest UTS that reached 89% of the UTS of BM. It is about a 25% improvement compared to laser weld. Further increase in heat input is unfavorable because it widens the tempered zone which leads to the decrease in UTS. The degree of softening is independent of the heat input. The cause of the higher microhardness of FZ of hybrid welds will be further investigated.

## Figures and Tables

**Figure 1 materials-14-03943-f001:**
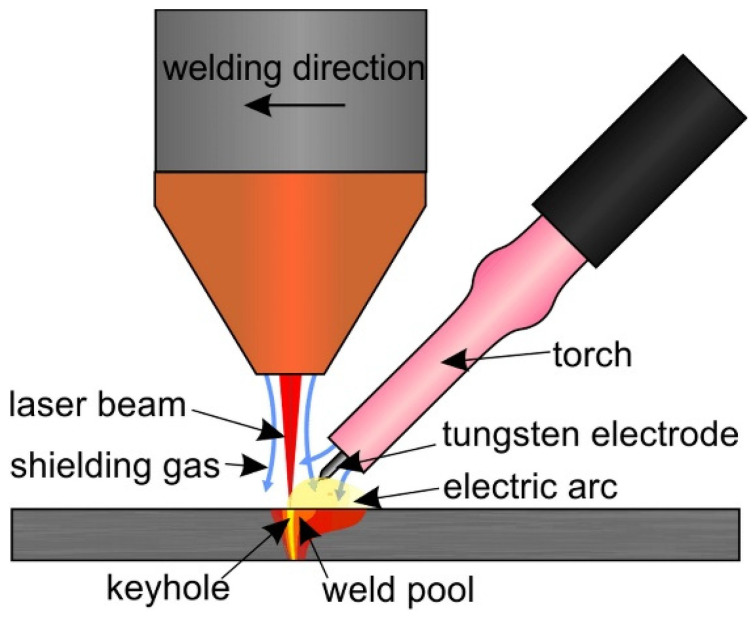
Experimental set-up.

**Figure 2 materials-14-03943-f002:**
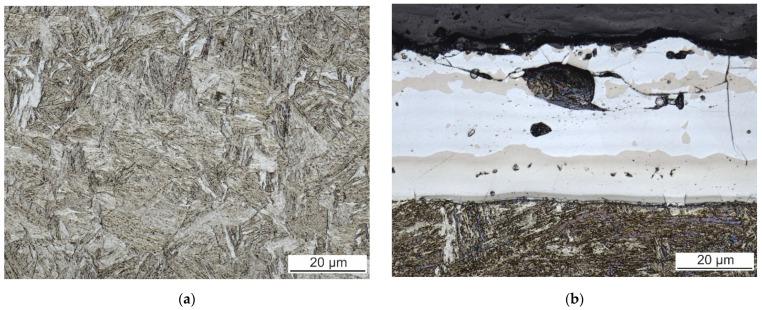
(**a**) Microstructure of the BM. (**b**) Microstructure of surface coating.

**Figure 3 materials-14-03943-f003:**
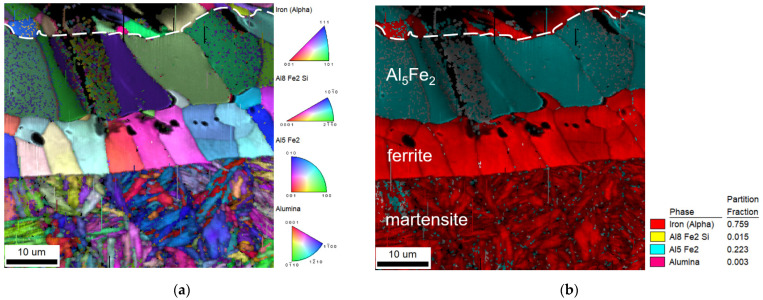
(**a**) IPF and (**b**) corresponding phase and IQ map of the coating–steel interface.

**Figure 4 materials-14-03943-f004:**
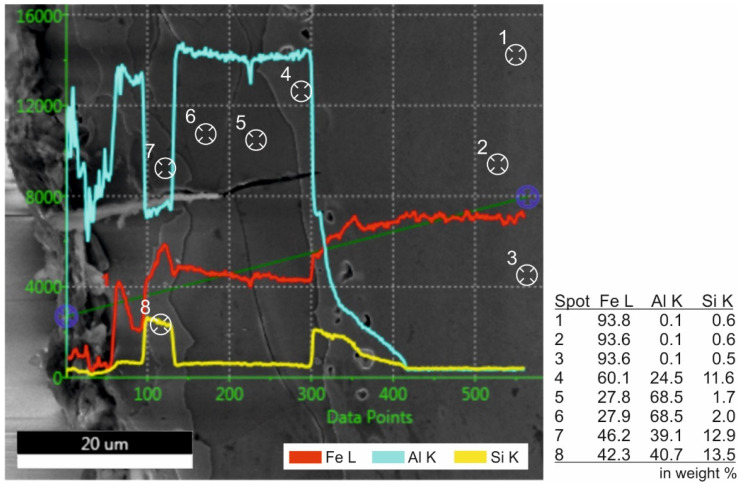
The EDS analysis of surface coating and BM.

**Figure 5 materials-14-03943-f005:**
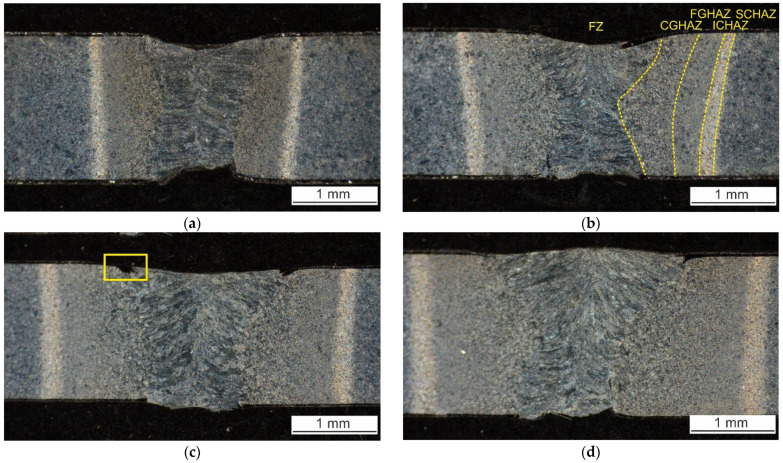
The macrostructure of (**a**) laser weld and hybrid welds made with (**b**) 20 A, (**c**) 40 A, and (**d**) 60 A.

**Figure 6 materials-14-03943-f006:**
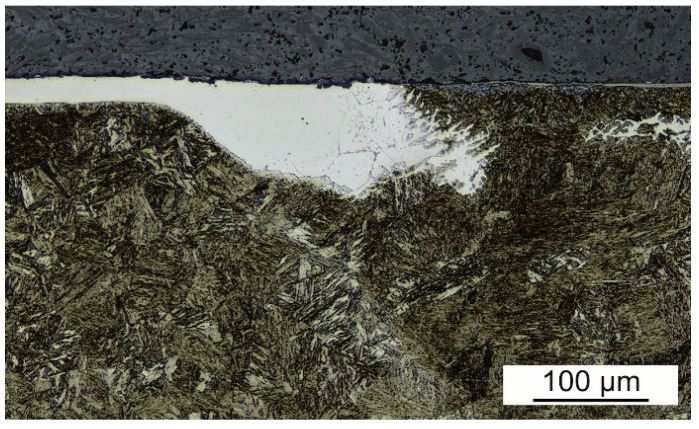
Surface area of hybrid weld made with 40 A (signed with a yellow rectangle in Figure 5).

**Figure 7 materials-14-03943-f007:**
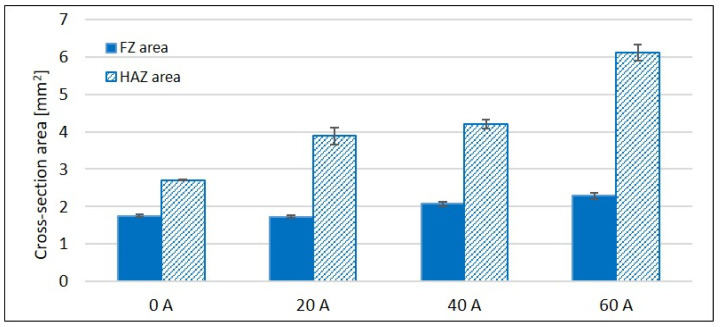
Weld cross-section area.

**Figure 8 materials-14-03943-f008:**
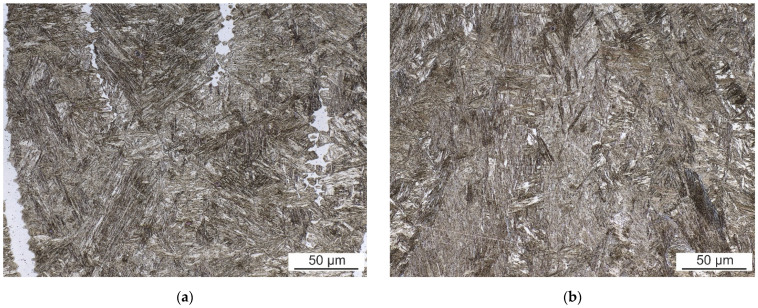
FZ of (**a**) laser weld and (**b**) hybrid weld made with 60 A.

**Figure 9 materials-14-03943-f009:**
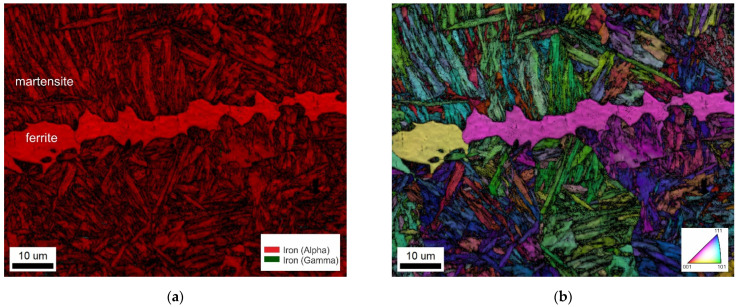
(**a**) Phase map and (**b**) IPF+IQ map of laser weld FZ.

**Figure 10 materials-14-03943-f010:**
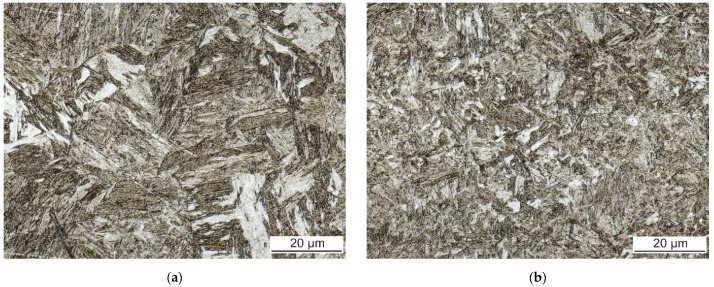
Subzones of HAZ of hybrid weld made with 20 A: (**a**) CGHAZ, (**b**) FGHAZ, (**c**) ICHAZ, and (**d**) SCHAZ.

**Figure 11 materials-14-03943-f011:**
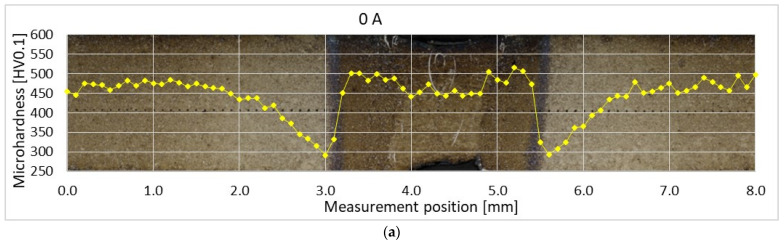
Microhardness of (**a**) laser weld and hybrid welds made with (**b**) 20 A, (**c**) 40 A, and (**d**) 60 A.

**Figure 12 materials-14-03943-f012:**
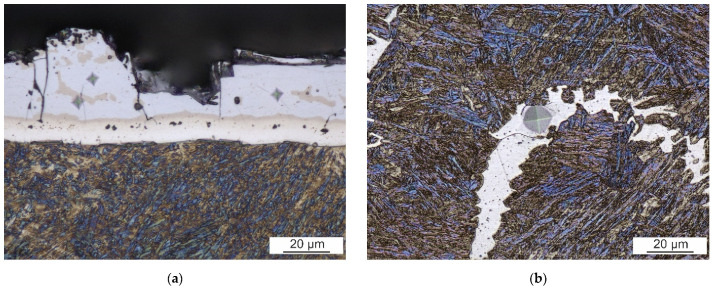
Microindents (**a**) in the surface coating and (**b**) in the ferritic region of FZ of a laser weld.

**Figure 13 materials-14-03943-f013:**
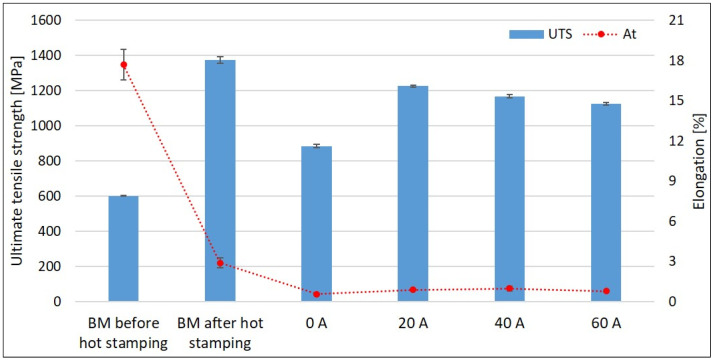
Tensile properties of the BM and welded samples.

**Table 1 materials-14-03943-t001:** Chemical composition of 22MnB5 measured by GDOES in weight percent.

C (%)	Mn (%)	Si (%)	P (%)	S (%)	Cr (%)	Ni (%)	Cu (%)	Al (%)	Ti (%)	Sn (%)	B (%)	Fe (%)
0.22	1.13	0.20	0.015	0.001	0.19	0.03	0.03	0.032	0.01	0.01	0.0032	bal.

## Data Availability

The data presented in this study are available on request from the corresponding author.

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
