# Peer review of "Microstructural Characterization of Laser Weld of Hot-Stamped Al-Si Coated 22MnB5 and Modification of Weld Properties by Hybrid Welding"

_materials, 2021, doi:10.3390/ma14143943_

Round 1

Reviewer 1 Report

Comments:

  1. Introduction - the effect of boron on hardenability should be supported by literature references
  2. Line 158 - original microstructure - as-received state
  3. Line 168-170 - the description of EBSD parameters should be included in the Methods section.
  4. Fig. 3b. It was found that the coating only consists of the Al5Fe2 phase, which seems to be incorrect (what happened to the silicon - spot 7and 8 - Fig. 4?) 
  5. Line 175-177 - the description of EDS analysis should be included in the Methods section.
  6. Fig. 7. The analysis of FZ and HAZ zones based on the etching of welded joints can be treated as an estimate (it is not accurate), the range of standard deviation(?) also seems to be larger - then the trend could be determined - linear/exponential.
  7. Fig. 10. "Therefore, it looks almost similar to the BM." - there are significant differences resulting from the severe tempering - precipitation processes.
  8. Why was such a small load used in the hardness distributions (HV0.1)? This reduces the reliability of the results.

Author Response

Dear Reviewer,

Thank you for your helpful comments and advice. We did our best to improve our manuscript based on them. Here are our responses:

ad1) Two references were added.

ad2) Corrected.

ad3) Moved

ad4) The data is correct. The Al5Fe2 is the widest but not the only phase of the coating after hot stamping. The spots 4, 7, and 8 belong to the silicon-rich (12-14% Si) ferritic zones that surround the Al5Fe2 band. The solubility of silicon in alpha-iron is up to about 18 %. Thus, almost all silicon is embedded in the ferritic solid solution. It is also mentioned in the text now.

ad5) Moved.

ad6) We agree that it is just an estimate. However, we believe that it is sufficient for the visualization of the effect of heat input. The SCHAZ and BM look similar on macrographs, their interface is not eye-visible. Therefore, the HAZ area was measured only across the CGHAZ, FGHAZ, and ICHAZ. Weld dimensions fluctuate along the weld length. Three cross-sections of each weld were analyzed. The average value and standard deviations were calculated. We added this information into the text and re-wrote the sentence about the trend.

ad7) We apologize for this incorrect statement. It has already been fixed.

ad8) The small load was chosen for detailed characterization of each subzone of the weld (especially in the HAZ) at the same depth. A larger load would result in larger indents. Therefore, the larger spacing would be required to meet the requirements of ISO 6507-1. This would result in a less detailed description of each subzone. According to ISO 9015-2, a load of 0.98 N is normal for the measurement of microhardness on the weld cross-section.

Reviewer 2 Report

The paper presents a series of information on laser welding and hybrid welding of Al-Si coated 22MnB5 steel.

From the analysis of the information presented in the article, I found the following:

- The paper presents a series of results that are of little interest to the scientific community:

- The introductory part needs to be substantially improved taking into account other bibliographic sources. Also, at the end of the Introduction sections, a detailed structure of the paper and a presentation of the research objectives must be presented;

- Better characterization is required in terms of mechanical properties for Al-Si coated 22MnB5;

- Additional information on the technology of production and the dimensions of the tensile test samples must be added;

- A series of completions are required regarding the technological parameters of the welding process (especially for TIG);

- The decision to make hybrid welds with 20A must be explained; also for the parameters 40A; 60A;

- The discussion part needs to be improved as the novelty of the research results compared to other current results in the field is not highlighted in this section;

- Conclusions should be more concrete and include future research directions.

Author Response

Dear Reviewer,

Thank you for your helpful comments and advice. We have tried to improve our manuscript based on them. Here are our responses:

  • The introduction was improved. More references were added as well as more detailed objectives are presented.
  • Concrete values of mechanical properties were added.
  • The production technology of tensile test specimens was added. More details about their dimensions is in subchapter 2.2 now.
  • We believe that laser welding was sufficiently described by laser power, focal point position, spot diameter, welding speed, and pressure of shielding gas. Concerning TIG welding, we added the welding regime beside the current description (diameter of tungsten electrode and its position towards the beam, welding current, welding speed, and pressure of shielding gas) in subchapter 2.1.
  • The current 20 A was the minimum current at which a stable arc was achieved at 20 mm/s, which is a relatively high welding speed for TIG welding. The experiment was designed to evaluate the effect of increasing current on mechanical properties of welded samples, especially the tensile strength. The values above 60 A represent the arc supply energy larger than that of the laser beam and lead to undesirable widening of HAZ. This information was added at the end of subchapter 2.1.
  • Both the discussion a conclusions were improved.

Reviewer 3 Report

This research dealt with the laser-TIG hybrid welding for HPFed steel. Molten pool flow can be enhanced by arc heat source, which is much stronger than that of induced by laser keyhole movement. Maybe additional arc heat source ensures the cooling rate, and made a uniform element distribution inside molten pool. It can be inferred that hybrid process is effective process manage the impurities. 

  1. As the authors noted, an additional heat source (TIG) affects the cooling rate. However, fusion zones in Fig. 11 (a) and (b) have a lower Vickers hardness values compared to (c) and (d). Please comment that reason.
  2. Probably, the reason for the low welding speed can be deduced to maintain the arc stability. The process speed is limited by the arc (any kinds of MIG and TIG) when the hybrid process is applied. It is recommended that remove the mentioning of productivity in the introduction part. 20 mm/s of welding speed is too slow in the laser welding process.
  3. HPF steel is commonly applied for the bumpers and pillars in automotive. Welding is performed before heat treatment as like TWB joint in most of cases. The author performed the laser-arc hybrid welding on heat-treated HPFed steel. Please mention the industrial target.

Author Response

Dear Reviewer,

Thank you for your helpful comments and advice. We have tried to improve our manuscript based on them. Here are our responses:

ad1) The cooling rate is decreased by the additional heat of the arc. Therefore, one could expect lower hardness of hybrid welds compared to laser weld. However, it is not only about the cooling rate (that is high above the critical quenching rate). Also, the chemical composition of FZ can vary because a different volume of coating (proportional to the FZ width) is dissolved in the steel at different heat inputs. The portion of FZ width and FZ cross-section area increases with increasing heat input. Therefore, a higher content of silicon can be expected in the FZ at higher heat inputs. Silicon strengthens the ferrite and increases the hardenability of austenite. Therefore, higher content of silicon can be responsible for the FZ hardness increase. This might explain our results. Nevertheless, these assumptions must be confirmed yet. These thoughts have been added to the discussion.

ad2) Welding speed is limited also by laser power, required penetration depth, and positioning system. The welding speed of 20 mm/s is typical for our system with the 2 kW laser. We did not have to reduce it when the arc was employed. The 5-axis systems or 6-axis robots are usually used for laser welding in automotive. Then, the maximal welding speed is limited to about 100 mm/s. Higher welding speeds (up to about 400 mm/s) can be applied only in the case of welding in 2D using positioning portals. You are true that the arc stability would not be maintained at such high processing speeds. Although the speed of laser welding was maintained in our experiments, we rather accept your recommendation.

ad3) The typical applications of 22MnB5 are bumper beams, door impact reinforcements, A-pillar reinforcement, B-pillar reinforcement, floor reinforcements, tunnel reinforcement, dash panel cross members, and roof cross members. These hot-pressed parts must be welded to produce a complex shape of body-in-white. It is not possible to produce such complex shapes (e.g. A-pillar + roof cross members + B pillar reinforcement) in a single step of pressing. In addition, the protective coating of the sheets is locally destroyed after welding. This would lead to oxidation during subsequent hot forming.

Round 2

Reviewer 2 Report

The authors revised their manuscript according to my suggestions. Thus the manuscript can be accepted for publication